# Proposal of a Quantitative Assessment Method for Viewpoint Geosites

**Marco Túlio Mendonça Diniz** [1,]* and **Isa Gabriela Delgado de Araújo** [2]

1 Department of Geography, Seridó Higher Education Centre, Federal University of Rio Grande do Norte, Caicó 59300-000, Brazil
2 Graduate Program and Research in Geography, Centre for Humanities, Literature and Arts, Federal University of Rio Grande do Norte, Caicó 59300-000, Brazil
* Correspondence: tulio.diniz@ufrn.br

**Abstract:** The evaluation of viewpoint geosites is a recent topic in geosciences, as most works deal with a more general analysis of places and areas, but this one deals with something more specific. Therefore, the general objective of this paper is to propose a method for evaluating viewpoints, based on the assumption that it is necessary to use scientific and aesthetic values as core values in quantitative evaluation. The method used was built based on criteria from other authors, relating the issue of viewpoints to geodiversity, considering scientific and aesthetic values as central, but in addition to other values. With the application of this method at some viewpoints in Rio Grande do Norte, Brazil, it was possible to verify that from 12 sites, 9 were considered geomorphosites and only 3 geodiversity sites. Consequently, this method shows a significant response in highlighting the potential of a site, its geological composition, geomorphology, and landscape visualisation.

**Keywords:** scientific; aesthetic; potentiality; geology; geomorphology

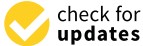



## 1. Introduction

Studies concerning geodiversity have highlighted many different, innovative perspectives and new methods are being developed to preserve and conserve the abiotic environment. Gray stated that geodiversity comprises "the natural range (diversity) of geological (rocks, minerals, fossils), geomorphological (landforms, topography, physical processes), soil and hydrological features. It includes their assemblages, structures, systems and contributions to landscapes" [1] (p. 12).

Several methods have been proposed to evaluate geodiversity in a qualitative and quantitative scope, either in a broad (considering scientific, economic, aesthetic, tourism, and other values) or restricted way (considering the scientific value as being central). Some works developed for the evaluation of geological and geomorphological heritage can be cited, such as those by Cendrero [2], Panizza [3], Brilha [4], Pralong [5], Reynard [6], Pereira [7], Reynard et al. [8], Lima [9], Garcia-Cortéz and Urqui [10], Pereira [11], Brilha [12], and Reynard et al. [13].

Pereira [7] introduced the evaluation of viewpoints in the landscape as potential extrinsic character indicators. Viewpoints were identified as a specific type of geosite in the work of Fuertes-Gutiérrez and Fernández-Martínez [14]; these authors classified geosites into five categories: points, sections, areas, complex areas, and viewpoints.

Recently, Migoń and Pijet-Migoń [15] (p. 512) defined viewpoints as "localities which offer a wider look at the surrounding landscape and hence, better understanding of its history, spatial relationships between rock types and landform categories (i.e., geodiversity), and ongoing environmental change". It can be seen that viewpoints are still a recent topic within the geosciences and more specific studies on the topic are lacking.

Viewpoint sites demonstrate an important relationship between geoheritage and the environment for three reasons. The first is that sites should not only focus on geology

and geomorphology but also on landscape visualisation. The second reason is that environmental conditions can affect the geological parameters of the viewpoint, making the visibility of abiotic elements better or worse. The third reason is that the value of these sites is closely related to their aesthetic attributes and the very act of viewing is a basic aesthetic judgment [16].

Among other works, it was observed in Araújo [17] and Diniz et al. [18] that viewpoints are underestimated when evaluating geomorphological heritage. The viewpoints evaluated in these works, although intuitively considered to be exceptional areas from an aesthetic and scientific point of view, fell into the categories of "places of geomorphological interest" or "geodiversity sites," since the evaluation method used did not contemplate the area visualised from the viewpoints, but the sites themselves.

A problem arises because of this: which method should be employed in order to evaluate the viewpoints?

This paper aims to propose a method for evaluating viewpoint sites, considering that scientific and aesthetic values must be used as core values in quantitative evaluation.

The research is justified due to the scarcity of proper methods for quantifying viewpoint geomorphological sites in the worldwide literature. These are exceptional sites, where the heritage to be inventoried and valued is not found in the site itself (in situ) but in the landscape that can be viewed from it (ex situ).

## 2. Materials and Methods

The sites were characterised and quantified using the method proposed by the authors, in order to specifically evaluate the viewpoints. This method was developed at some viewpoints in the state of Rio Grande do Norte, Brazil.

### 2.1. Study Area

For the application of the proposed method, we selected viewpoints in three distinct geomorphological contexts in the state of Rio Grande do Norte, Brazil. The first context was two crystalline plateaus covered by flat-top sandstones; the second context was a residual hill in a marginal sedimentary basin; and the third context was a coastal environment with inactive cliffs (resulting from Quaternary tectonics) and moving dunes. The viewpoints are presented in Figure 1.

The plateaus of Martins-Portalegre and João do Vale present the same geomorphological context, because there is a topographic inversion in both areas, through two processes: the differential erosion of the basement and the syn- and post-rift tectonics of the Potiguar marginal sedimentary basin (or Potiguar Basin). The first process was responsible for the exhumation of intracrustal structures, such as batholiths, from transcurrent shear zones. The second refers to the reactivation associated with uplift pulses commanded by Cenozoic magmatism, which contributed to the uplift of the crystalline massifs covered by the Serra dos Martins Formation sandstones at high altitudes [19].

Therefore, according to Maia et al. [19], this area exposes a lateritic sandstone capping, which covers both the plateaus of Martins and Portalegre, and João do Vale; the laterite is substantial enough for the maintenance of flat tops. Both have a lithological resistance associated with the Itaporanga and Poço da Cruz intrusive suites. What differs between them is the rainfall data [20]. The Martins region has average annual rates of 1106.8 mm, while João do Vale is within an isohyetal zone of 700–800 mm.

The Serra de Mossoró is located in the buffer zone of the Furna Feia National Park, and integrates the Potiguar Basin, presenting itself as a residual relief formed by sedimentary sandstone and limestone rocks that presented facies of greater resistance to erosion. The presence of lateritic sandstone at its apex reaches a maximum altimetry of 268 m, which differs from its surroundings (flattened relief), with altimetric quotas of around 100 m. These sedimentary units correspond to the Cretaceous (Jandaíra Formation) and the Neogene (Barreiras Formation), respectively [21].

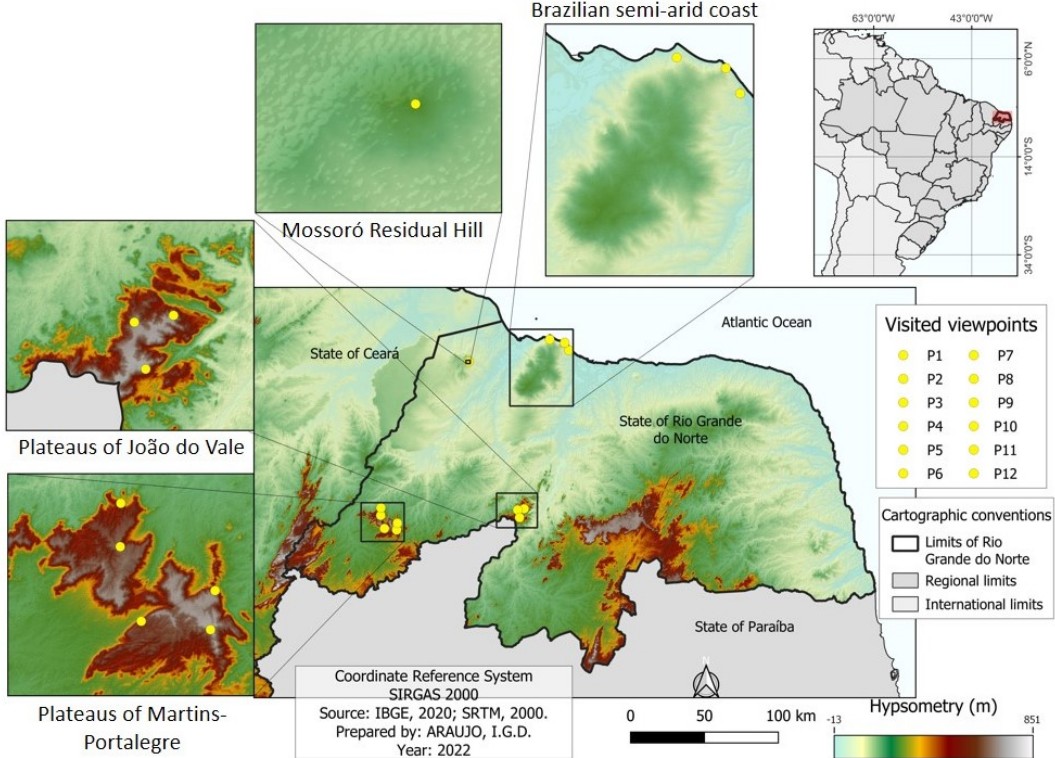

**Figure 1.** Viewpoint Location Map. Key: P1—Sunset Viewpoint of Diadema; P2—"Viewpoint under construction"; P3—Tota's viewpoint; P4—Boa Vista viewpoint; P5—Ponta da Serra view-point; P6—Novo Mundo viewpoint; P7—Sunset Viewpoint; P8—Muriçoca viewpoint; P9—Serra viewpoint; P10—São Cristóvão Dunes; P11—Mirante das Crosses; P12—Rosado Dunes. Source: Elaborated by the authors.

The coastal points are located in the most arid area of the Brazilian coast, called Costa Branca (the 'White Coast' in English). The term Costa Branca refers to the fact that, on this coast, the forms tend to have a whitish colour and include the marine beaches, the mobile dunes, and the largest sea salt industry on the continent, located on naturally hypersaline plains. The Costa Branca is located in the passive border context which predominates the entire Atlantic coast of South America. These are stable coastlines, from a tectonic point of view, dominated by sedimentary deposition processes and relief flattening to form incredibly low topography with average altitudes of less than 30 m. However, the area surrounding Serra do Mel is undergoing active Quaternary tectonics. The dome of Serra do Mel was raised to an altitude of 280 m in its central part and this elevation has had repercussions on the coast, as the Ponta do Mel, a cliff with more than 120 m of altitude (now inactive, after a period of continuous uplift), and a decrease in mean sea level over the last 2100 years has caused levels of marine terraces to appear between the base of the cliff and the coastline. The forms and processes of the area in question come from Cenozoic tectonics; climatic oscillations, common throughout the period after the last glacial maximum and their concomitant eustatic oscillations; marine transgressions and regressions; as well as from dissection processes found in the landscape [22,23].

Figures 2–4 represent the investigated locations of different geomorphological contexts.

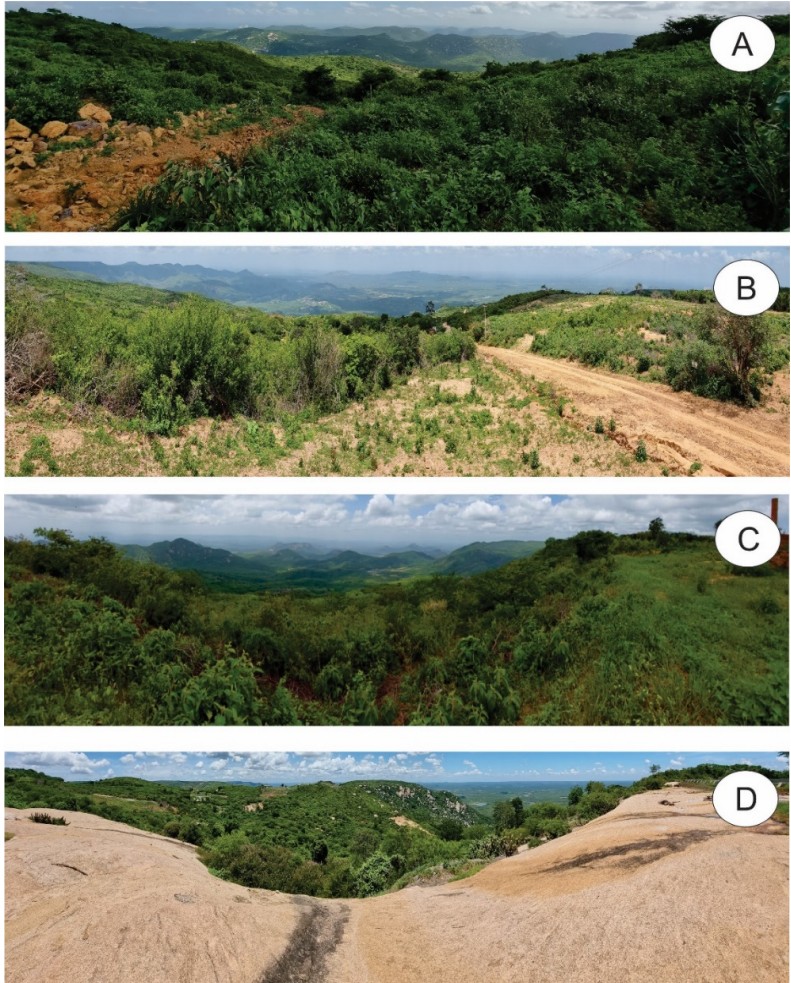

**Figure 2.** Viewpoints in the context of the João do Vale massif, which features the Serra dos Martins Formation—(**A**) Muriçoca Viewpoint, (**B**) Sunset Viewpoint, (**C**) Novo Mundo Viewpoint, and (**D**) Ponta da Serra Viewpoint. Source: Authors' collection (2022).

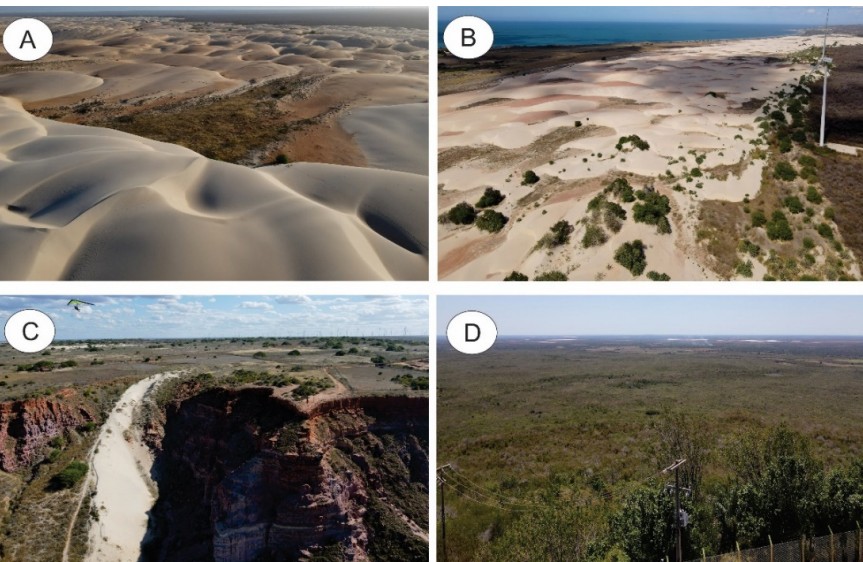

**Figure 3.** Viewpoints in the context of the Brazilian Semi-arid Coast and the witness hill. Locations: (**A**) Rosado Dunes, (**B**) São Cristóvão Dunes, (**C**) Ponta do Mel Viewpoint, and (**D**) Mossoró Ridge. Source: Authors' collection (2022).

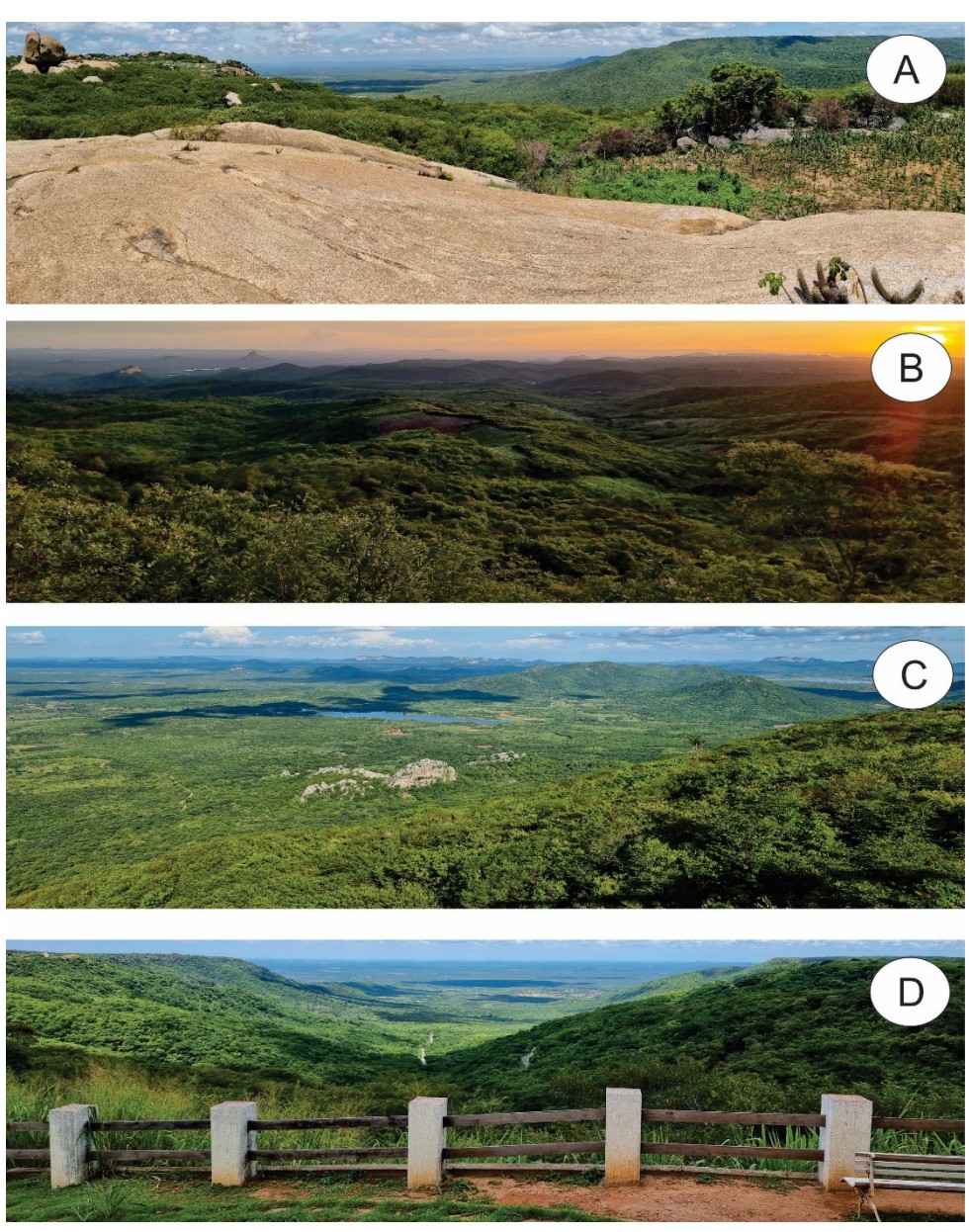

**Figure 4.** Viewpoints in the context of the formation massif of the Martins Mountains. Locations: (**A**) Tota Viewpoint, (**B**) Diadema Sunset Viewpoint, (**C**) Viewpoint "under Construction" and (**D**) Viewpoint Boa Vista. Source: Authors' Collection (2022).

### 2.2. The Methods

There are geoconservation steps for sites of abiotic interest to be preserved and conserved. According to Brilha [4], the inventory and quantification steps are the first to be carried out. These are studies of a qualitative and quantitative nature, applicable to geodiversity, using primary sources (by means of observations and descriptions) and secondary sources (through information carried out in the geosciences). They all use bibliographic survey techniques, field research, and data validation, which are essential for the categorisation of research objects.

For the quantitative evaluation, a method was proposed to specifically evaluate the geosite viewpoints. The main characteristics of geodiversity were systematised with the main elements of the viewpoints. The studies by Mikhailenko and Ruban [16], Mikhailenko et al. [24], and Kubalíková et al. [25] presented elements directly related to the object and were incorporated, in part, in the proposed method.

Mikhailenko and Ruban [16] evaluated 17 criteria in a semi-quantitative way, with scores from 0 to 4, and these criteria are defined as:

- Presence of geological elements;
- Presence of geoheritage elements;
- Presence of geomorphological elements;
- Presence of geomorphological heritage elements;
- Presence and type of vegetation cover;
- Presence of uncovered soil;
- Presence of water (rivers, lakes, and seas);
- Presence of snow/ice;
- Presence of cultural/historical elements (e.g., historical buildings);
- Degree of anthropogenic intervention (stress);
- Degree of landscape richness (number of different types of elements mentioned above);
- Degree of landscape degradation (because of natural processes such as rockfalls or anthropogenic processes such as massive construction, land abandonment, etc.);
- Degree of landscape cleanness (absence of garbage left by residents and/or tourists);
- Degree of landscape openness;
- Degree of landscape fragmentation (mosaic);
- Degree of landscape contrast (visual difference of landscape—for instance, by colour or by height of landscape elements); and
- Degree of visible component details belonging to landscape elements (this depends on both the distance to observable environments and the size of elements).

Mikhailenko et al. [24] considered seven criteria for the semi-quantitative evaluation of viewpoints that are based on and observed from bridges. This method considered panoramas and other views, visibility of unique geological/geomorphological features, diversity of visible unique geological/geomorphological features, accessibility, special constructions for comfortable observation, the geological value of the bridge itself, and the cultural value of the bridge itself. Kubalíková et al. [25] conducted an evaluation regarding geoeducation and geotourism, with the purpose of applying a pilot study on viewpoint studies so that, within this framework, one could develop active management and specific activities. Reynard [6], Reynard et al. [8], and Reynard et al. [13] presented the most widely used methods in the literature for the evaluation of geomorphological heritage. This method was also considered by Mucivuna et al. [26]. In this way, geodiversity values (scientific, aesthetic, and additional values) were evaluated, considering the characteristics of the points of view [16,24], in addition to some of the methods proposed by Pereira [11], Brilha [12] and Kubalíková et al. [25].

Kirillova et al. [27], when writing about what makes a destination beautiful to visitors, identified that tourists generally judge colourful and vibrant landscapes as beautiful. The same authors identified that visitors are attracted to more diverse landscapes and the findings were incorporated into the method through criterion B5. Kirillova et al. [27] also identified that destinations with unique characteristics are seen as being beautiful and, therefore, have a pull on visitors. The proposed method incorporates this dimension through the rarity criterion, which was adapted from Diniz et al. [18] (B7). Rarity is also present in the works by Reynard [6], Reynard et al. [8], Reynard et al. [13], and Brilha [12], among others.

The proposed method for the quantitative evaluation of opinions considers scientific (Table 1), aesthetic (Table 2), and additional (tourist, cultural and didactic, Table 3) values. All values are realised by summation, with parameters ranging from 1 to 4 for each criterion. Very low values are considered to be <25% of the total value, low values are between 25% and 50% of the total, medium values are between 50% and 75% of the total, and high values are 75% or more of the total realisable value of the score.

**Table 1.** Scientific value of the points of view.

| Core Values—Visualised Landscape | | | |
|---|---|---|---|
| **Scientific Value** | | | |
| **Criteria** | **Definition** | **Parameters** | **Score** |
| **A1—Diversity of visible geological/geomorphological features (forms and processes)** | Amount of geological/geomorphological elements visible in the visualised landscape. | 1–4 viewable elements | 1 |
| | | 5–7 viewable elements | 2 |
| | | 8–9 viewable elements | 3 |
| | | $\geq$10 viewable elements | 4 |
| **A2—Representativeness** | Indicates the relevance of the site as a record of elements or processes related to the geomorphological evolution of the region and the context in which it is inserted, as well as the use of geomorphology for society. | Absence of any relevant aspect of scientific nature. | 1 |
| | | It contains illustrative records of elements or processes of geodiversity without expressiveness. | 2 |
| | | It contains illustrative elements that represent type sections of formations or are used as classic examples of geomorphological elements or processes or land use for society. | 3 |
| | | Containing a geoform that represents a classic form and processes of landscape evolution, the viewpoint allows unique aspects to be seen within a 200 km radius. | 4 |
| **A3—Integrity** | Indicates the conservation level of the viewable area and the possibility of viewing aspects of interest. | Altered observable area, the visualisation of aspects of interest is quite restricted, with no possibility of being easily retrieved. | 1 |
| | | Observable area altered, but still allows visualisation of aspects of interest with the possibility of recovery. | 2 |
| | | Observable area with some anthropic alteration, but human occupation does not limit the visualisation of the features of interest. | 3 |
| | | Observable area preserved without the need for reclamation or human use in no way affects the visualisation of aspects of geomorphological interest. | 4 |
| **A4—Paleogeographic Value** | The importance of the object for the reconstruction of the Earth's climate and history (e.g., Cenozoic tectonic relief) is evaluated by this criterion. | It contains illustrative elements that represent paleogeographic evolution but shows anthropic alteration or the presence of vegetation. | 1 |
| | | It contains illustrative elements that represent the paleogeographic evolution, without the presence of alteration and vegetation cover, allowing excellent visualisation of the geomorphological elements. | 2 |
| | | A significant area for local paleogeographic understanding can be viewed. | 3 |
| | | A key area for understanding regional paleogeographic evolution can be visualised. | 4 |
| **Classification** | | | |
| Very low | | | 1–4 |
| Low | | | 5–8 |
| Medium | | | 9–12 |
| High | | | 13–16 |

Source: [6,8,11–13,17,18].

**Table 2.** Aesthetic value of the points of view.

| Core Values—Visualised Landscape | | | |
|---|---|---|---|
| **Aesthetic Value** | | | |
| **Criteria** | **Definition** | **Parameters** | **Score** |
| **B1—Overview** | Angle from which you can observe the landscape | Restricted views on one or two sides | **0** |
| | | 120–180° panorama from one side only | **1** |
| | | 120–180° panorama on one side and restricted view on the other side. | **2** |
| | | 120–180° two-sided panoramas. | **3** |
| | | 360° Panorama | **4** |
| **B2—Visibility of the geological/geomorphological characteristics of the landscape** | Elements visualised in the landscape | Poor (very general view, presents obstacles such as vegetation, massifs, etc.). | **1** |
| | | Mixed (some features are more visible than others). | **2** |
| | | - | **-** |
| | | Excellent (all details are visible). | **4** |
| **B3—Verticality** | Height at which the viewpoint is located | Flat or gently undulating viewpoint. | **1** |
| | | Viewpoint on a strongly undulating relief. | **2** |
| | | Viewpoint on a residual hill or inselberg. | **3** |
| | | Viewpoint on an escarpment. | **4** |
| **B4—Presence of water bodies** | Existence of water in the landscape | Absence of water bodies | **1** |
| | | - | **-** |
| | | Lakes and/or Rivers | **2** |
| | | - | **-** |
| | | Ocean | **4** |
| **B5—Colour contrast and individual elements** | Contrasting colours from the RGB of an icnographic document and the presence of individual elements, such as an inselberg. Homogeneous landscape—composed of few and mostly similar elements. Heterogeneous landscape—composed of a complex configuration of very diverse elements, many contrasting colours and/or vibrant colours in the landscape | Homogeneous landscape without individual elements. | **1** |
| | | Homogeneous landscape with up to three individual elements. | **2** |
| | | 3–5 contrasting colours or heterogeneous landscape. | **3** |
| | | Contrast of 6 or more colours and heterogeneous landscape and/or vibrant colours in the landscape. | **4** |
| **B6—Visualisable area (km$^2$)** | Area where you can observe the landscape from the viewpoint. | <50 km$^2$ | **1** |
| | | 50 < 300 km$^2$ | **2** |
| | | 300 < 500 km$^2$ | **3** |
| | | >500 km$^2$ | **4** |

**Table 2.** *Cont.*

| Core Values—Visualised Landscape | | | |
|---|---|---|---|
| **Aesthetic Value** | | | |
| **Criteria** | **Definition** | **Parameters** | **Score** |
| **Rarity** | Importance of the area visualised from the site in terms of its geomorphological occurrence in the investigated area | Visualised area of common occurrence in the study area, between 6 and 10 formations with similar characteristics can be viewed in the area, within the same geomorphological context within a radius of 200 km. | 1 |
| | | Up to 5 formations with similar characteristics can be viewed in the area, within the same geomorphological context within a 200 km radius. | 2 |
| | | Up to 3 formations with similar characteristics can be viewed in the area, within the same geomorphological context within a 200 km radius. | 3 |
| | | A unique formation can be viewed in the area within a 200 km radius or $\geq 3$ within a 500 km radius. | 4 |
| **Classification** | | | |
| **Very low** | | | 1–7 |
| **Low** | | | 8–14 |
| **Medium** | | | 15–21 |
| **High** | | | 22–28 |

Source: [6,8,11–13,16–18,24,25].

**Table 3.** Additional Viewpoint Values.

| Supplementary Values—Own Site | | | |
|---|---|---|---|
| **Touristic Value** | | | |
| **Criteria** | **Definition** | **Parameters** | **Score** |
| **C1—Accessibility** | Indicative of difficulties in accessing the site. | Only by prepared pedestrians (e.g., in the case of suspension bridges). | 1 |
| | | By pedestrians only. | 2 |
| | | By cars with dirt roads. | 3 |
| | | By cars with paved roads. | 4 |
| **C2—Tourism category** | The existing tourism purposes in the area (sun and beach, geotourism, ecotourism, adventure, studies, sports, fishing, cultural, religious, etc.). | 0–1 type of tourism | 1 |
| | | 2 types of tourism | 2 |
| | | 3 types of tourism | 3 |
| | | 4 or more types of tourism | 4 |
| **C3—Existence of use in progress** | Indicates the current conditions of tourist use of the site. | Site without any current use or site with some visitation rate, but still incipient. | 1 |
| | | Site with average visitation rate and presence of accommodation. | 2 |
| | | Site with a high rate of visitation but without a mechanism to control visitors and with the presence of accommodation. | 3 |
| | | Site with a high rate of visitation and equipped with measures and the presence of lodging facilities less than 3 km away. | 4 |

**Table 3.** *Cont.*

| Supplementary Values—Own Site | | | |
|---|---|---|---|
| **Touristic Value** | | | |
| **Criteria** | **Definition** | **Parameters** | **Score** |
| **C4—Convenience** | Pleasant built environment with the presence of bars, restaurants, inns, internet, banks, among others. | 0–1 convenience element. | 1 |
| | | 2 convenience elements. | 2 |
| | | 3–4 convenience elements. | 3 |
| | | ≥5 convenience elements. | 4 |
| **C5—Signaling** | Signs as a means of communication for tourists | Absence of signage. | 1 |
| | | Presence of identification plates, indicative and informative signs about the risks of the site. | 2 |
| | | Presence of indicative signs of abiotic relevance. | 3 |
| | | Interpretive panels of the area. | 4 |
| **C6—Safety** | Condition of being safe in place. Presence of fences, chest guards, warning signs about the exposed dangers, among others. | A steep or non-steep viewpoint with no protection for the visitor. | 1 |
| | | A viewpoint with 1 protection element for the visitor. | 2 |
| | | Viewpoint with more than 2 visitor protection elements. | 3 |
| | | Steep viewpoint (less than or greater than 45°) with more than 3 visitor protection elements. | 4 |
| **Cultural Value** | | | |
| **Criteria** | **Definition** | **Parameters** | **Score** |
| **D1—cultural relevance** | It illustrates the site's association with cultural elements. Use for religious purposes, place names, or holding cultural events. | No connection with cultural elements. | 1 |
| | | Indirect and direct relationship with cultural elements (ruins, place names, cave paintings) and/or craft activities. | 2 |
| | | Site with the presence of some cultural element which makes an ancillary contribution to the visit or use of the site. | 3 |
| | | Close relationship with cultural elements (cultural landscape), where the cultural aspect is one of the main attractions of the area. | 4 |
| **Didactic Value** | | | |
| **Criteria** | **Definition** | **Parameters** | **Score** |
| **E1—Didactic relevance** | Potential of the site to illustrate geodiversity elements or processes and the possibility of using the site for teaching geosciences by schools. | It can be used for didactic purposes in graduate studies. | 1 |
| | | It can be used for didactic purposes for undergraduate students. | 2 |
| | | It can be used for teaching purposes for high school students. | 3 |
| | | It can be used for didactic purposes for the general public or elementary school students. | 4 |
| **Classification** | | | |
| **Very low** | | | **1–8** |
| **Low** | | | **9–16** |
| **Medium** | | | **17–24** |
| **High** | | | **25–32** |

Source: [11,16,17,24].

The proposed quantitative evaluation takes into consideration parameters that are most related to scientific and aesthetic values but also considers tourism/management, cultural, and didactic values as well.

An inventory of the sites was also conducted based on the inventory sheet proposed by Araújo [17].

Sites with high scientific value (>75%) and/or high aesthetic value (>75%) were considered viewpoint geosites. If the sites did not have these values, they were designated as geodiversity sites when they had high or medium scores in the additional values and/or medium scores in the core values.

## 3. Results

A quantitative evaluation was carried out for 12 viewpoints in the state of Rio Grande do Norte, Brazil. Of the sites evaluated and analysed, nine were considered to be geosites; they are: Ponta da Serra viewpoints (high scientific value), Rosado Dunes (high scientific and aesthetic value), Ponta do Mel (high scientific and aesthetic value), "Em Construção" (high scientific and aesthetic value), Sunset (high scientific value), Serra from Mossoró (high scientific and aesthetic value), Muriçoca (high scientific value), Sunset Viewpoint of Diadema (high scientific value) and Dunas de São Cristóvão (high aesthetic value). The viewpoints Tota's, Novo Mundo, and Boa Vista were considered to be geodiversity sites, presenting medium values in the evaluation. In terms of scientific value, the highest values presented in Table 4 were assigned to the viewpoints Ponta da Serra (16), Rosado Dunes (16), Ponta do Mel (16), Sunset (14), Serra de Mossoró (14), Muriçoca (14), Sunset Viewpoint of Diadema (14), and "Under Construction" (13). Only the viewpoints Tota's (12), Novo Mundo (12), Boa Vista (12), and São Cristóvão Dunes (11) achieved a medium score.

**Table 4.** Scientific value of viewpoints.

| Sites | Scientific Value | | | | |
|---|---|---|---|---|---|
| | **A1** | **A2** | **A3** | **A4** | **Total** |
| Ponta da Serra Viewpoint (Portalegre) | 4 | 4 | 4 | 4 | **16** |
| Rosado Dunes—Porto do Mangue | 4 | 4 | 4 | 4 | **16** |
| Ponta do Mel Viewpoint | 4 | 4 | 4 | 4 | **16** |
| Sunset Viewpoint—Triunfo Potiguar | 2 | 4 | 4 | 4 | **14** |
| Serra de Mossoró Viewpoint | 2 | 4 | 4 | 4 | **14** |
| Muriçoca Viewpoint—Jucurutu | 4 | 4 | 4 | 2 | **14** |
| Sunset Viewpoint of Diadema—Martins | 3 | 3 | 4 | 4 | **14** |
| "Viewpoint Under Construction"—Martins | 2 | 4 | 4 | 3 | **13** |
| Tota's Viewpoint—Serrinha dos Pintos | 2 | 3 | 4 | 3 | **12** |
| Novo Mundo viewpoint—Jucurutu | 2 | 3 | 4 | 3 | **12** |
| Boa Vista Viewpoint—Portalegre | 2 | 3 | 4 | 3 | **12** |
| São Cristóvão Dunes—Areia Branca | 2 | 3 | 3 | 3 | **11** |
| **Classification** | | | | | |
| Very low | | | | 1–4 | |
| Low | | | | 5–8 | |
| Medium | | | | 9–12 | |
| High | | | | 13–16 | |

Legend: A1—Diversity of visible geological/geomorphological features (forms and processes); A2—Representativeness; A3—Integrity, and A4—Paleogeographic value. Source: Prepared by the authors (2022).

Five sites had a high aesthetic value, as can be seen in Table 5; they are the viewpoints: Serra de Mossoró (26), Ponta do Mel (24), Rosado Dunes (23), "Under Construction" (22), and São Cristóvão Dunes (22). The average scoring sites were Ponta da Serra (20), Muriçoca (20), Miradouros do Tota (18), Pôr do Sol (18), Miradouro Pôr do Sol de Diadema (18), Novo Mundo (17), and Boa Vista (16).

**Table 5.** Aesthetic Value of Viewpoints.

| Sites | Aesthetic Value | | | | | | | |
|---|---|---|---|---|---|---|---|---|
| | **B1** | **B2** | **B3** | **B4** | **B5** | **B6** | **B7** | **Total** |
| Serra de Mossoró Viewpoint | 4 | 4 | 3 | 4 | 3 | 4 | 4 | 26 |
| Ponta do Mel Viewpoint | 3 | 4 | 4 | 4 | 4 | 1 | 4 | 24 |
| Rosado Dunes—Porto do Mangue | 4 | 4 | 2 | 4 | 4 | 1 | 4 | 23 |
| "Viewpoint Under Construction"—Martins | 2 | 4 | 4 | 1 | 3 | 4 | 4 | 22 |
| São Cristóvão Dunes—Areia Branca | 4 | 4 | 2 | 4 | 3 | 1 | 4 | 22 |
| Ponta da Serra Viewpoint (Portalegre) | 4 | 4 | 2 | 1 | 3 | 4 | 2 | 20 |
| Muriçoca Viewpoint—Jucurutu | 2 | 4 | 2 | 1 | 3 | 4 | 4 | 20 |
| Tota's Viewpoint—Serrinha dos Pintos | 2 | 4 | 2 | 1 | 3 | 4 | 2 | 18 |
| Sunset Viewpoint—Triunfo Potiguar | 2 | 4 | 2 | 1 | 3 | 4 | 2 | 18 |
| Sunset Viewpoint of Diadema—Martins | 1 | 4 | 4 | 1 | 3 | 1 | 4 | 18 |
| Novo Mundo Viewpoint—Jucurutu | 2 | 4 | 2 | 1 | 3 | 3 | 2 | 17 |
| Boa Vista Viewpoint—Portalegre | 1 | 4 | 4 | 1 | 3 | 1 | 2 | 16 |
| **Classification** | | | | | | | | |
| **Very low** | | | | 1–7 | | | | |
| **Low** | | | | 8–14 | | | | |
| **Medium** | | | | 15–21 | | | | |
| **High** | | | | 22–28 | | | | |

Legend: B1—Panorama and other views; B2—Visibility of geological/geomorphological features; B3—Verticality; B4—Presence of water bodies; B5—Colour contrast and individual elements, and B6—Viewable area (km$^2$). Source: Prepared by the authors (2022).

For the additional values, only the viewpoint Ponta do Mel (25) showed a high value; medium scores were presented at the viewpoints Boa Vista (21), Rosado Dunes (21), Tota (20), Sunset (19), Serra de Mossoró (17), and Pôr-do-Sol de Diadema (17). Five sites scored low values, namely the viewpoints São Cristóvão Dunes (16), Novo Mundo (15), Ponta da Serra (13), "Under construction" (13), and Muriçoca (13), as seen in Table 6.

**Table 6.** Additional Values Results from Viewpoints.

| Sites | Additional Values | | | | | | | | |
|---|---|---|---|---|---|---|---|---|---|
| | C | | | | | | D | E | Total |
| | **C1** | **C2** | **C3** | **C4** | **C5** | **C6** | **D1** | **E1** | |
| Ponta do Mel Viewpoint | 3 | 3 | 2 | 3 | 3 | 3 | 4 | 4 | 25 |
| Boa Vista Viewpoint—Portalegre | 3 | 2 | 4 | 4 | 1 | 2 | 1 | 4 | 21 |
| Rosado Dunes—Porto do Mangue | 4 | 2 | 2 | 3 | 2 | 1 | 4 | 3 | 21 |
| Tota's Viewpoint—Serrinha dos Pintos | 3 | 2 | 3 | 3 | 1 | 1 | 3 | 4 | 20 |
| Sunset Viewpoint—Triunfo Potiguar | 3 | 3 | 1 | 2 | 1 | 2 | 3 | 4 | 19 |
| Serra de Mossoró Viewpoint | 4 | 2 | 3 | 2 | 1 | 1 | 0 | 4 | 17 |
| Sunset Viewpoint of Diadema—Martins | 4 | 2 | 2 | 1 | 1 | 2 | 1 | 4 | 17 |
| São Cristóvão Dunes—Areia Branca | 4 | 1 | 1 | 3 | 1 | 1 | 1 | 4 | 16 |
| Novo Mundo Viewpoint—Jucurutu | 3 | 1 | 1 | 1 | 1 | 1 | 3 | 4 | 15 |
| Ponta da Serra Viewpoint (Portalegre) | 3 | 1 | 1 | 1 | 1 | 1 | 1 | 4 | 13 |
| "Viewpoint under construction"—Martins | 3 | 2 | 1 | 2 | 1 | 1 | 1 | 2 | 13 |
| Muriçoca Viewpoint—Jucurutu | 3 | 1 | 1 | 1 | 1 | 1 | 1 | 4 | 13 |
| **Classification** | | | | | | | | | |
| **Very low** | | | | | 1–8 | | | | |
| **Low** | | | | | 9–16 | | | | |
| **Medium** | | | | | 17–24 | | | | |
| **High** | | | | | 25–32 | | | | |

Legend: C (TOURIST VALUE): C1—Accessibility; C2—Tourist category; C3—Existence in current use; C4—Convenience; C5—Signalling; C6—Safety; D (CULTURAL VALUE): D1—Cultural Relevance; and E (DIDACTIVE VALUE): E1—Didactic Relevance. Source: Prepared by the authors (2022).

## 4. Discussion

The construction of this method provides a new way to evaluate the geomorphological heritage of viewpoints, in order to attribute essential characteristics to the realities of these sites without underestimating their potential for geoconservation. In this way, the observation, analysis, and evaluation are not only of the site itself but also of the viewpoint, i.e., what is visualised in the landscape.

In the works of Araújo [17] and Diniz et al. [18], it was observed that several geomorphological elements in the evaluation were underestimated, undervaluing the existing potential that can be representative of society, mainly due to the different dimensions and characteristics of other places.

The studies by Mikhailenko and Ruban [16], Mikhailenko et al. [24], and Kubalíková et al. [25] were important steps for this type of evaluation; however, a way of linking the values of geodiversity with the central elements was barely considered for geosite definition, as portrayed by Brilha [12], Reynard et al. [13], and Diniz et al. [18]. Of these, Brilha and Reynard et al. considered the scientific value to be central, while Diniz et al. considered the scientific and aesthetic values to be central.

In the proposed method, the central values are only considered to be in the visualised area, as this is the main object of interest in viewpoint studies. Additional values include both the site (in situ) and the observed area (ex situ).

In the central values, all criteria refer to the visualised area and not to the site itself. In the additional values, there are criteria that evaluate the site itself, such as accessibility, the existence of use in progress, and safety.

The Ponta da Serra, Dunas do Rosado, Ponta do Mel, Sunset Viewpoint, Serra de Mossoró, and Muriçoca Viewpoint are fundamental sites for regional palaeogeographic understanding, which contributed decisively to the high scores of these sites. The method was considered efficient in capturing the points of view with the greatest potential for exploring scientific value. The Ponta do Mel viewpoint, for example, is the only area with tectonic relief on the Brazilian coast, where there is a Quaternary uplift that is still active (Figure 5). The base of the cliff has laterite and higher marine terrace levels from the foot of the cliff and is already inclined due to the influence of the active uplift of the area. These terraces are followed by terraced levels of lower elevation, which are less inclined as one approaches the ocean. The dynamics of the atmosphere can also be explored in the area, as the sea breezes ascend in the canyon, which allows for free flight. The image in Figure 5 shows hang gliders flying in the area.

The Rosado Dunes, seen in Figure 6, represent the only sample of red dunes in Brazil. The colouring is a result of the provenance of marine sediments which were added to the sediments of the Rosado cliffs. These cliffs are also inactive and formed by post-barrier unconsolidated sedimentation. Since the last marine transgression, these sediments were uplifted by the reactivation of the Afonso Bezerra fault in the area. In this 360° viewpoint, there are several records of the history of the earth over the last three thousand years.

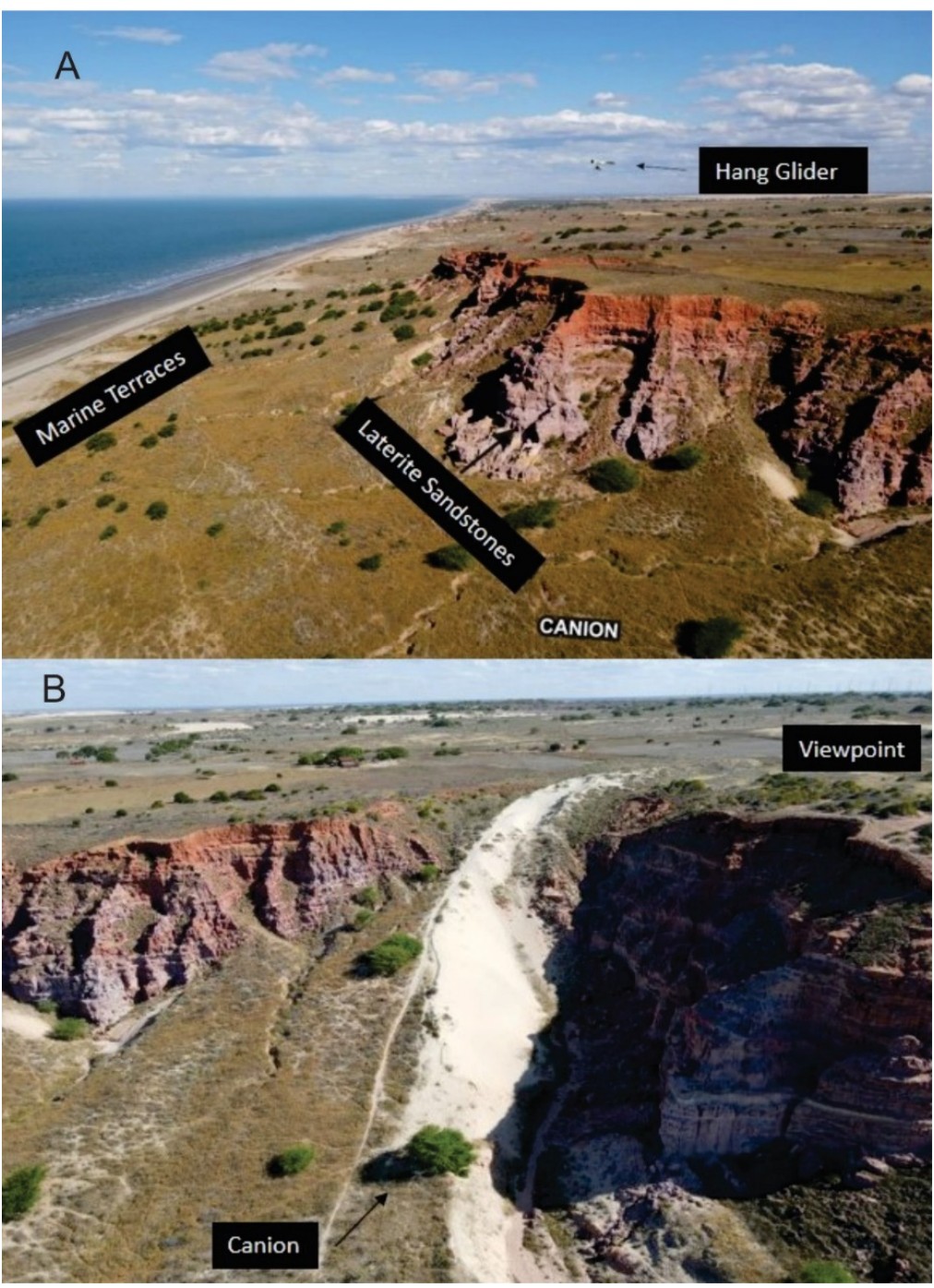

**Figure 5.** Serra do Mel's cliff. (**A**) view from the viewpoint and (**B**) an image of the area from where the canion can be seen partly covered by eolian paleodune sediments that entered the canion during the Holocene carried by the sea breeze channeled into the canyon. Source: Collection of the authors (2022).

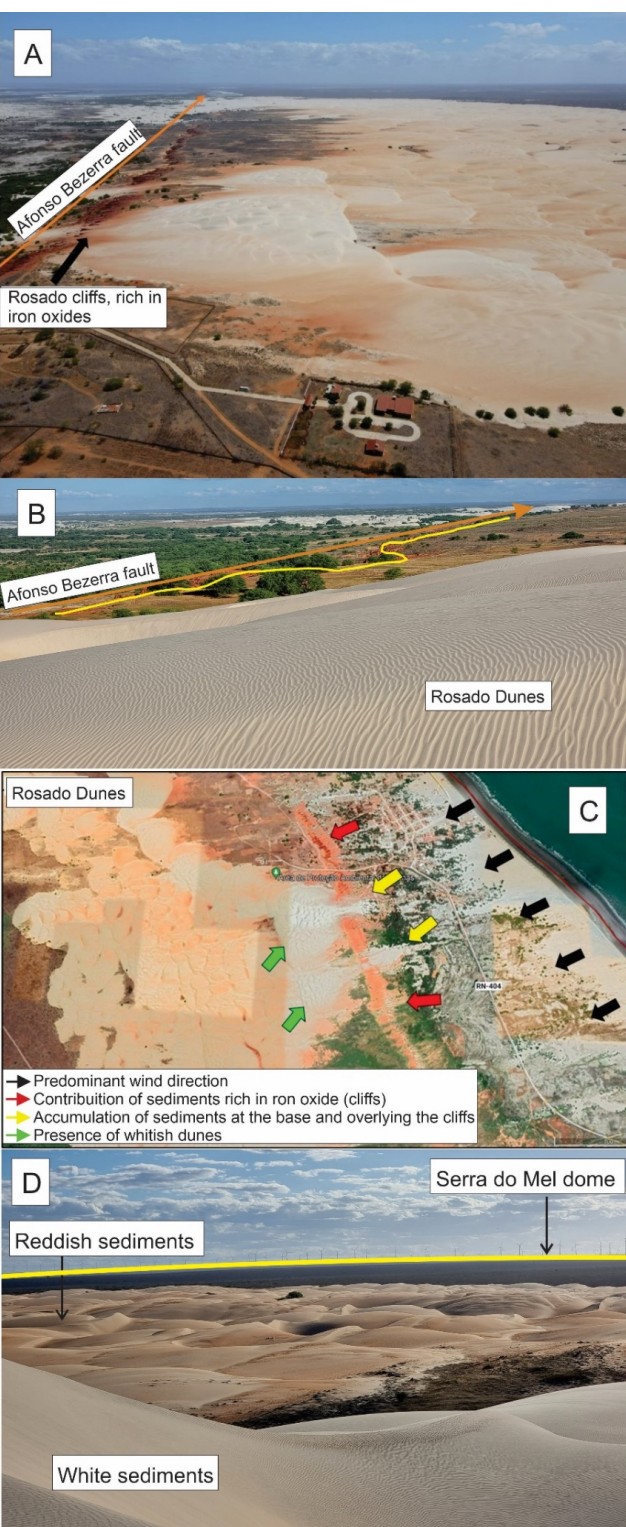

**Figure 6.** The mosaic images represent a passive margin relief with divergent plates, which undergo neotectonics and have become the only active structure on the South American continent. (**A**) Aerial image of the dunes and cliffs of Rosado, highlighting the direction of the Afonso Bezerra fault; (**B**) Image from the Rosado Dunes viewpoint, from where the source cliff of the reddish sediments controlled by the fault can be seen. (**C**) Satellite image, illustrating the main processes in the area. (**D**) Image from the Rosado Dunes viewpoint from where the Serra do Mel dome and the white and reddish sediments of the dunes can be seen. Source: (**B**,**D**) Collection of the authors (2022); (**A**,**C**) adapted with permission Borges [23].

The area of the Dunas do Rosado geosite neighbours the Rosado cliffs and also has great cultural value: some Brazilian movie scenes and Netflix series have been filmed there.

In terms of aesthetic value, the viewpoints with the highest scores were Serra de Mossoró, Ponta do Mel, and Rosado Dunes. All three sites showed important geological and geomorphological features, rarity, six-colour contrast, and a visible area of more than 500 km$^2$ (in the case of Serra de Mossoró) and vibrant colours (in the cases of Ponta do Mel and Rosado Dunes).

The Serra de Mossoró viewpoint, for example, is a residual isolated hill, with 360° views of other high reliefs, set in a radius of at least 100 km of the flattened sandstone and limestone relief of the Potiguar Basin. The area is occupied by constructions such as communication antennas, and the construction of a 3 or 4 m high lookout on top of this peak would provide a 360° panoramic view for visitors. The cost of construction would be low, but this could boost the tourism that is already practiced in the area, with restaurants that explore the lookout at a slightly lower level than the top of this relief.

Figure 7 shows hundreds of km$^2$ of relief that was flattened during the Quaternary, around the Serra de Mossoró. The landscape is extremely complex with natural vegetation and fruit plantations (melons are grown for export). This image shows the Atlantic Ocean (more than 35 km away from the lookout) and even the port-island of Areia Branca, an artificial island that functions as a port for the export of the area's salt production; the port is about 55 km away from the lookout. As the viewpoint is 360° and the hill is in extremely isolated relief, it is possible to see the dome of the Serra do Mel, the Mossoró River valley, the crystalline plateaus of the interior of the state, and the salt production in the nearby estuaries. It is certainly one of the most privileged points to observe geomorphological forms and processes at great distances; the method was effective in capturing the high aesthetic value of the main areas where the method was tested.

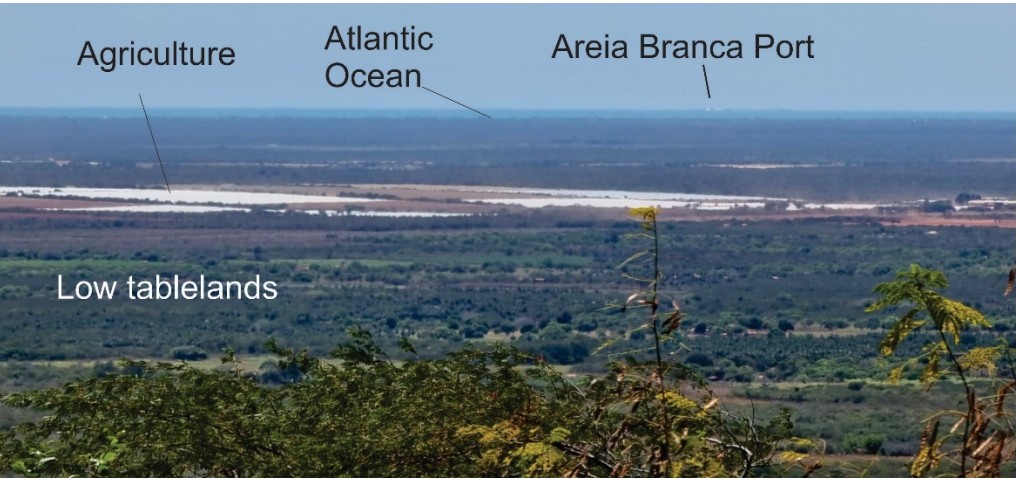

**Figure 7.** Serra de Mossoró Viewpoint. Source: Collection of the authors (2022).

The aesthetic values varied much more between sites, mainly due to the difference in the proportion of viewable area; higher sites with more isolated reliefs allow wider views and are likely to score higher on this value. However, this criterion can be offset by the rarity criteria, since sites with restricted vision can provide the visualisation of extremely rare forms in the context studied, as in the case of Ponta do Mel.

Figure 8 shows a representation of two types of geosites, Serra de Mossoró and Ponta do Mel, with different dimensions: the first has a 360° view, while the second represents a more restricted observation. The Serra de Mossoró geosite, with a large observable area, obtained the highest score in terms of aesthetic value, while the Ponta do Mel geosite, with a more restricted view, scored low in criterion B6 but maintained its high score in aesthetic value, due to rarity (B7) of the observable area from the viewpoint.

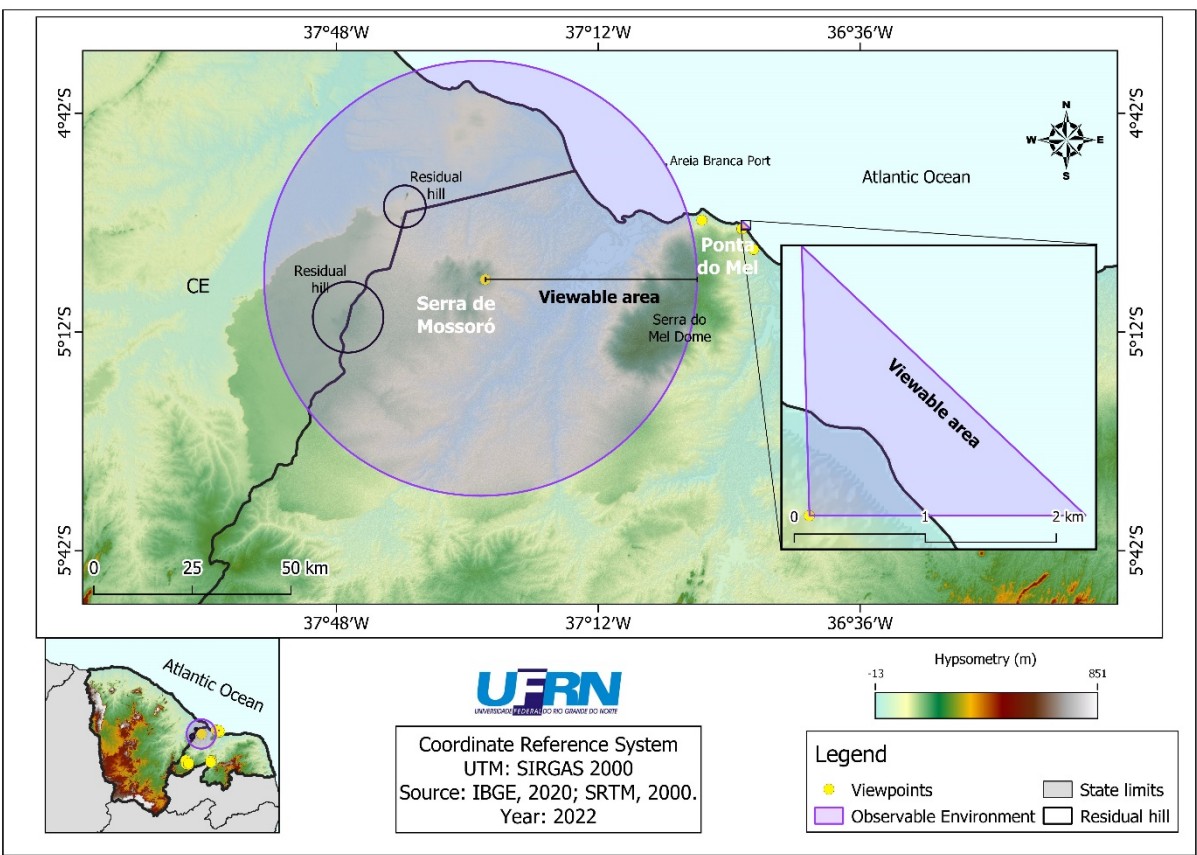

**Figure 8.** Representation of viewpoint geosite types. Source: Elaborated by the authors.

The Dunas de São Cristóvão viewpoint did not have a high scientific value, but it can be considered a geosite for its high aesthetic value, which is due to the vibrant colours of the extremely heterogeneous landscape that can be seen in the area. The viewpoint has a 360° view, excellent visibility, and provides a view of the Atlantic Ocean. The dunes follow a flow of bypass parallel to the coastline; this current flow is due to the predominant E-W wind direction, which is the same as the predominant direction of the coastline, as seen in Figure 9.

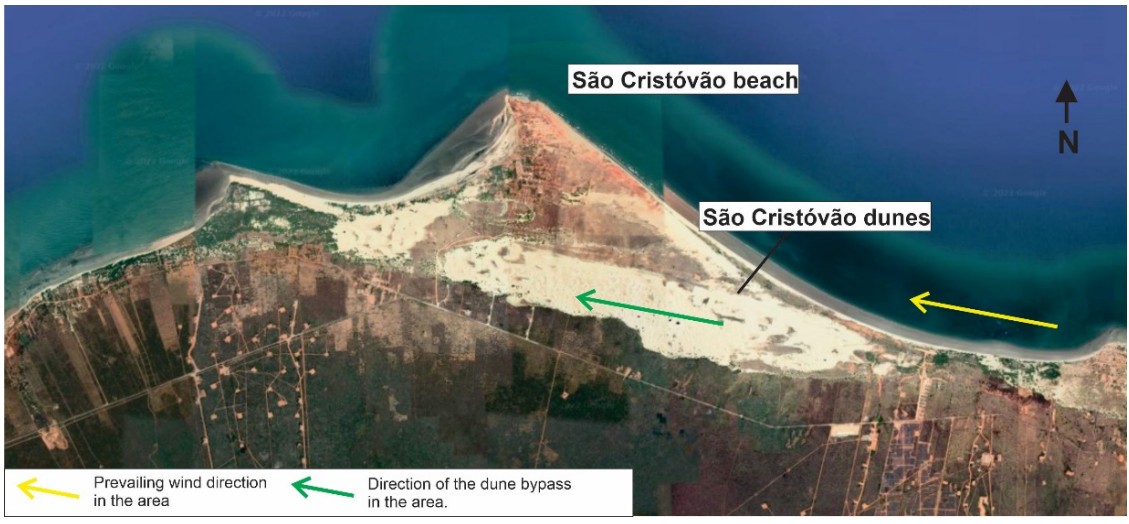

**Figure 9.** São Cristóvão Dunes area. Source: Google Earth (Data SIO, NOAA, U.S. Navy, NGA, GEBCO).

For the additional values, only one site had a high rating, due to the fact that some viewpoints still do not have tourist infrastructure that can draw the attention of visitors. The geosites that have been noted have a lot of geological/geomorphological potential that needs to be publicised to favour not only tourists, but the local community, helping to increase income in this sustainable tourism segment.

## 5. Conclusions

The theme of viewpoints is relatively recent in geodiversity studies. In most of the evaluation methods used to date, for both geological and geomorphological heritage, these sites were underestimated. Thus, a specific quantification is needed to address their particularities and highlight their value to society, attracting visitors in general, so that the sites can be used in a sustainable and profitable way for the local community.

From the proposed method, it was possible to consider eight geosites in terms of scientific value. Four of these also obtained a high aesthetic value: Serra de Mossoró, Ponta do Mel, Dunas do Rosado, and a Viewpoint "under construction". Dunas de São Cristóvão was considered a geosite only for its aesthetic value, which shows the importance of considering aesthetics in the evaluation, as it exposes the degree of visibility of the geomorphological elements, as well as the rarity of the observable environment (in a regional or local configuration). The aesthetic value of a site has a high power to attract visitors to the area to be conserved.

The method developed in this paper has shown a positive result for viewpoint evaluation, highlighting its main elements, besides what is seen in the landscape, and exposing a specific focus, which has not been seen in other proposals. It is applicable to any area with a natural viewpoint.

**Author Contributions:** Conceptualization, I.G.D.d.A. and M.T.M.D.; methodology I.G.D.d.A. and M.T.M.D.; software, I.G.D.d.A. and M.T.M.D.; validation, I.G.D.d.A. and M.T.M.D.; writing—original draft preparation, I.G.D.d.A.; writing—review and editing, I.G.D.d.A. and M.T.M.D.; funding acquisition, M.T.M.D. All authors have read and agreed to the published version of the manuscript.

**Funding:** This research was funded by the Coordination for the Improvement of Higher Education Personnel (CAPES), the Research Support Foundation of Rio Grande do Norte (FAPERN) and National Council for Scientific and Technological Development (CNPq), and "The APC was funded by Federal University of Rio Grande do Norte".

**Acknowledgments:** The authors thank the National Council for Scientific and Technological Development (CNPq) for funding field activities and the Research Productivity grant (for the first author) and the doctoral scholarship from the Coordination for the Improvement of Higher Education Personnel (CAPES) (for the second author). The authors thank The Research Support Foundation of Rio Grande do Norte (FAPERN) for funding field activities.

**Conflicts of Interest:** The authors declare no conflict of interest.

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
