# Peer review of "Proposal of a Quantitative Assessment Method for Viewpoint Geosites"

_resources, doi:10.3390/resources11120115_

Round 1
Reviewer 1 Report
The paper by Diniz and Araújo is an interesting attempt to apply semi-quantitative approach to the evaluation of viewpoint geosites, using examples from north-east Brazil. As the topic is a relevant one and viewpoint geosites are indeed potential resources for local tourism development (perhaps more appealing to the general public than ‘classic’ geosites), exploration of the theme is justified and I do not have objections regarding possible publication in ‘Resources’.
Having said that, however, I also note that the paper requires considerable further work before it can be offered to the global scientific community and become, hopefully, a valuable source of reference for geoheritage, geoconservation and tourism specialists and practitioners. My critical comments are of two kinds: one group touches the core of the paper, which is the evaluation method itself. Other comments are pertinent to the structure of the paper and the way of presentation.
The following issues emerge from the ‘Methods’ section:
· first and MOST IMPORTANTLY, I see no justification of the proposed numerical scoring, which seems rather inconsistent. For example, maximum score for diversity of geological elements is 30, whereas for representativeness is 50. Why? In some criteria we have three options to choose (geodiversity), in others – four (representativeness), in yet others – five (integrity). The same applies to two other sets of values. Such a loose approach does not build confidence in the method and some sort of unification/standardization has to be applied. Please note that this is NOT a minor remark that can be dismissed easily – this is KEY comment and I expect thoroughly different approach in the revised version if the paper is to meet standards of scientific presentation. I know it will require considerable work, but the alternative is to accept a proposal which in terms of scientific rigour, transparency and replicability is unsatisfactory.
· what is the purpose of repeating criteria used by Mikhailenko and Ruban (L163-183) if these are not used in this paper? This would be justified if the present authors show their own assessment of that scheme, perhaps leading to the conclusion that the approach hardly works (personally, I think the approach by Mikhailenko and Ruban is overcomplicated, in specific points too subjective, and difficult to apply in certain landscape contexts).
· the distinction between scientific, aesthetic and additional values is logical. Problems are with specific criteria and parameters, and these have to be better explained. For example, how do you understand “elements” in the criterion of diversity of geological/geomorphological characteristics? Given that rocks, structures and landforms show a certain hierarchy, this is a very important question. If I see two inselbergs from a viewpoint, are these two elements (two hills) or just one (landform category – inselberg)?
· in Representativeness, description of specific criteria raises doubts if this category is really about scientific representativeness of the panoramic view (why should use for the society matter?). In Integrity category, it is not clear what is assessed – the viewpoint itself or the area to be seen? If the latter, how do you decide on “deterioration” of the area? What are indicators of deterioration? Rewording of descriptions seems necessary. Similarly, description of palaeogeographical value should be disassociated from use/didactic potential, as we are still within the group of scientific values.
· for some criteria, numerical scoring only creates an impression of scientific rigour, whereas in reality the degree of subjectivity is very high (unless the authors are able to convince us that everything is clear-cut and unequivocal). This is evident from the description of parameters in the Aesthetic Value section, in the criteria of Visibility of geo-characteristics (how in practice should I distinguish between poor and mixed? or mixed and excellent?) and Colour contrasts. In the presence of water bodies, why lakes score 20 and artificial lakes score 30? Any justification?
· regarding Tourism category criterion, I am very sceptical if we can sensibly decide about different types of tourism practised at specific sites. These classifications (and which classification should we follow?) are our constructs, definitions are a matter of debate, whereas for the visitors themselves it is immaterial.
· in Safety criterion, the principle of evaluation should be whether the site is safe enough or not, not the number of protection elements. If one “visitor protection element” is sufficient to ensure safety, the site should receive the highest score. Quantity does not always mean quality.
· some criteria overlap. For example, in Representativeness didactic use is highlighted, whereas educational (didactic) potential is also included into Didactic relevance criterion.
· below zero values should not be introduced (Accessibility criterion) – scoring is not consistent already, but this makes the proposal even more chaotic and has no justification. Why the necessity of paying the fee (I guess “entrance free” is a typo, and it should be “entrance fee”) should reduce the tourist value of a site?
Summing up, the methodical approach requires considerable modifications, whereas its presentation in the paper should be expanded, mainly to explain how different parameters are understood and can applied. This is essential if the methodical approach is to be credible and replicable, as required by standards of science.
Comments regarding presentation:
· the distinction between transitional, observed and target environment after Mikhailenko and Ruban (L54-59 plus Fig. 1) is interesting, but is not considered in the evaluation of Brazilian sites in this paper and hence, the question about its relevance. Either make use of this distinction in discussion section or remove this paragraph and the figure.
· in L86-88 make it clear that coastal environment is ONE geomorphological context, with two different geomorphic features: inactive cliffs and mobile dunes.
· L94/Fig. 2 – two plateaus are mentioned by name, but these names are missing from Figure 2. Please add.
· recast L109-114. This is one, very long and convoluted sentence, difficult to understand.
· paragraph in L117-134 needs rewriting and simplification – in the current form it is not easy to understand.
· L135-137 are results, not part of “Study area”.
· Figures 3 to 5 in the present form are meaningless. Either expand the captions to tell the reader what geological/geomorphological features can be seen, or use annotations on the images themselves, and preferably BOTH.
· it would be very useful if 1-2 selected panoramic views (images) are described in detail, in terms of all criteria and parameters used in evaluation. This would increase the credibility of the evaluation proposal and help to relate views to the scores.
· most of text in section 3 “Results” duplicates the tables. Rather, expand explanation why the results are as they are.
· use annotations to enhance figures 6 and 7.
· I have doubts regarding figures 8 and 9 in a paper about viewpoint geosites, as it shows the landscape from an aerial perspective, inaccessible to a visitor.
· figure 10 is redundant.
· L370-372 – sounds a bit like wishful thinking…
· L381-382 – this is not necessarily so. I am afraid that some of the proposed criteria are site- or region-specific, e.g., those related to vegetation cover and water bodies (what about arid area viewpoint geosites?), cultural relevance (what about wilderness areas?), infrastructural development and convenience (standards vary from country to country, some sort of development is not permitted in protected areas). Recasting this part is recommended.
Finally, the language requires improvement in terms of wording, grammar and syntax. Some statements are difficult to understand because of language deficiencies.
Author Response
Dear reviewer, the following changes have been made:
- The method was changed according to the suggestions, both in the scope of the scores and modifications of criteria and parameters;
- The scoring was changed and was standardized from 1 to 4 by summation, being considered geosites those sites that presented >75% in the evaluation of the scientific and aesthetic values;
- The method tables have all been changed according to the new scores and classifications;
- The results and tables have all been changed according to the new scores;
- In the discussion, the paragraphs have been changed by changes in the results;
- The figure of the observable environment of the authors Mikhailenko and Ruban [16] was excluded;
- The central values in the evaluation agree with the viewable area;
- The teaching potential in the scientific value items have been changed and encompassed only in the Teaching Value, in the Additional Values;
- In the criterion of the presence of water bodies, there was a change regarding the issue of lakes;
- The zero score was excluded from the evaluation;
- Modification was made to the number of geomorphological contexts in the study area;
- The names of the plateaus were inserted in the location map;
- The figures became more explanatory about the study area;
- A grammatical revision was carried out with a specific company.
Reviewer 2 Report
Dear Authors,
Please see my comments enclosed in the attacjed document and in the followings:
Introduction: The authors summarise the problem correctly, the text is quite understandable. However, state-of-the-art is poorly described, the authors could have cited more relevant literature and could go deeper in definitions. I could imagine some kind of a geological and a geomorphological visualisation of the sample area.
Methods: quite well-developed methodology. Please be aware of good English language in the tables and their better visualisation.
Results: good results, but the wording is extremely bad: you use very long and difficult sentences that cannot be understood. Please, reshape the whole article with better English and better wording.
Conclusions: The first half is not a conclusion, but a summary of the proposed and solved problems. You could have written some summary data of the assessment. I miss the discussion.
Line 9: Look at the correspondence data
Line 12: informal wording
Line 25: English language issues
Line 30: Reference issue
Line 32: You are writing an article about geosite assessment - then why do you write here about geodiversity assessment? However, a more thorough overview on geosite assessment methods should be given.
Line 35: Reference issue
Lines 40-41: grammar issues
Line 43: grammar & reference issues
Line 46: grammar
Line 47: grammar & I do not really understand the second part of the sentence.
Line 63: reference
Line 64: bad wording
Line 70: delete then
Line 85: small initials please
Line 90: small initials please
Line 100: magnetism - magmatism?
Lines 109-114: extremely long sentence
Line 124: bad wording
Lines 124-130: extremely long sentence
Lines 150-155: the sentence is too long and complex to understand
Line 152: you write about geodiversity assessment here + there are much more relevant and newer articles from Brilha (eg. from the Geoheritage book published in 2018)
Line 157: grammar
Line 184: and
Line 185: delete 'on' & reword the sentence
Lines 191&192: what method was where considered? please reword the sentence
Line 194: viewpoint
Line 195: and
Line 198: stop the sentence here
Line 202: delete the hyphen
Line 205: the meanings of 'point of view' and 'viewpoint' are not the same!
Line 210: the tables are not too aesthetic
Line 224: Doest this site have a better name?
Line 236: This should be included in the table caption.
Line 244: small initials
Line 245: This should be included in the table caption.
Line 255: small initials
Lines 262-264: difficult to understand
Line 269: and
Lines 265-274: You have mentioned these aspects in the beginning of the article.
Line 275: bad grammar
Line 278: delete the
Line 283: small initial g
Line 293: delete 'there' & an open canyon can be observed
Line 296: stop
Line 299: figure?
Line 303: hyphen?
Line 306: stop
Line 307: stop
Lines 309-312: complex and wordy sentence
Lines 313-315: complex and wordy
Line 329: small initial
Lines 330-336: extremely long sentence
Lines 337-342: long sentence
Lines 350-355: long sentence

Author Response
Dear reviewer, the following changes have been made:
- Two important references have been added within the context of the article, the papers by Kubalíková, Kirchner and Kuda [25] and Kirillova et al. [26];
- Magnetism was changed to magmatism;
- The method tables have all been changed according to the new scores and classifications;
- As considerações finais foram alteradas;
- A grammatical revision was carried out with a specific company.
Reviewer 3 Report
Dear Authors,
This paper pays attention to an important and internationally urgent issue and makes serious advance in the understanding of viewpoint geosites on the basis of representative example. The paper is generally clear and well-organized. The methodology is strong, and the number of illustrations is big. Nonetheless, some improvements are necessary,
1) Key words: please, avoid the words from the title.
2) Figure 1 is taken from the original work. Can you elaborate it to avoid repetition?
3) Method: aesthetics are something more than you describe. Please, consider many aspects here: https://www.sciencedirect.com/science/article/pii/S0261517713002185
I do not claim to change your analysis, but, please, note that aesthetic properties are multiple. Also, please, check the works by Mikhailenko devoted to colours and patterns of geosites (see the issues of the open access journal “Geologos” – one paper was published in 2017 and the other in 2021).
4) In my opinion, Discussion should bear some implications for geoheritage and geotourism management. Note that the journal’s name is “Resources”, and, thus, you have to consider briefly the resource value of viewpoint geoheritage.
5) You demonstrate perfect awareness of developments, but, please, consider some works by Dr. Lucie Kubalikova. For instance, this one can be very useful: https://doi.mendelu.cz/artkey/doi-990001-9900_viewpoint-geosites-and-their-potential-for-geoeducation-and-geotourism.php
6) The writing is clear, but, please, polish it a bit else. For instance, I have not to capitalize the word “geodiversity”, and it should be spelled “paleogeographic” (not palaeo-geographic as in Table 1).
7) Please, avoid too short, one-sentence paragraphs.
Author Response
Dear reviewer, the following changes have been made:
- The keywords were modified, according to the suggestions;
- The figure of the observable environment of the authors Mikhailenko and Ruban [16] was excluded;
- Inserted the references of Kubalíková, Kirchner and Kuda [25] and Kirillova et al. [26] as suggested;
- The method tables have all been changed according to the new scores and classifications;
- A grammatical revision was carried out with a specific company.
Round 2
Reviewer 1 Report
L92 “basement” (not “embasement”)
L124-127 This is complicated sentence, difficult to follow. Recasting or splitting into two is recommended. The phrase “morphostructural genesis (elevation)” is ambiguous and we do not know what sort of process/mechanisms the authors imply.
Regarding the indicators and specific options for each I remain unconvinced that they can be easily apply and indeed, that all make sense (e.g., safety component – the site may not require any specific protection facilities, so low score is not necessarily evidence of poor suitability as a viewpoint). But I accept this as a proposal and let the readers decide whether they can apply this approach and will need to modify it.
L306 The phrase “there was a Quaternary uplift that is still active” is confusing. Either there WAS uplift or it IS still ongoing. You cannot have two at the same time. Please rephrase to capture the real situation properly. In the same line, it should be Figure 5 (not 6).
L307 I am not sure where the canyon is. We see an escarpment. Is it just one side of the canyon that we have a view at? If so, explain this in caption. Moreover, what makes you thinking that it is an inherited landform from “more humid paleoclimate”? Any solid evidence or just an assumption? If the latter, also make it clear in text.
L307-309 The sentence is not clear – consider rephrasing. “(…) base of the cliff is supported on laterite sandstone” – not sure what this means.
L319-327 To be honest, I have difficulties to follow this explanation and visualize the processes responsible for the current geomorphological landscape. What “flow parallel to the coastline” is implied in L324? Flow of what? Referring to L326, I am not sure what are these “several records”. Consider recasting this paragraph.
L450 I think that proper attribution of this paper is “Cendrero 2000” (this is the name under which the author is known). See no. 200 in his publication list available at https://personales.unican.es/cendrera/PubDic04.pdf. It also shows that the publication is in Spanish, not in English.
Table 1. Layout requires improvement. E.g., in the criterion A1 the name of the criterion applies to all first four rows, so the cells should be merged. The same is with Definition. In row 3 move Score to the right column. Look at the others criteria and make necessary adjustments. What is the meaning of brackets in the Score column (not everywhere present)? I do not think it adds anything and can be removed.
Tables 5 and 6 also need improvement of layout (uneven width of some rows).
Fig. 1. The main map should also include the trace of the coastline beoynd the borders of Rio Grande del Norte state. In the current layout, the coastal position is not evident.
Author Response
- L92 “basement” has been changed to basement, as suggested;
- L124-127 was remade
- L 306-319 was remade
- The paragraph about the flow parallel to the shoreline from the Rosado dunes was deleted and inserted into the discussions about the São Cristóvão dunes L 388-390;
- Another image has been inserted in Figure 5 in a mosaic format to highlight the canyon;
- Figure 9 was added to show the direction of the sediments in the dunes of São Cristóvão;
- Modified reference from Cendrero (2000).
- We consider that the tables are adequate and that formatting adjustments will be made by the publisher's final layout.